# Recent Progress of Metal–Air Batteries—A Mini Review

**Chunlian Wang [1], Yongchao Yu [2], Jiajia Niu [3], Yaxuan Liu [2], Denzel Bridges [2], Xianqiang Liu [3], Joshi Pooran [4], Yuefei Zhang [3] and Anming Hu [1,2,*]**

1 Institute of Laser Engineering, Beijing University of Technology, Beijing 100124, China
2 Department of Mechanical, Aerospace and Biomedical Engineering, University of Tennessee, Knoxville, TN 37996, USA
3 Institute of Microstructure and Properties of Advanced Materials, Beijing University of Technology, Beijing 100124, China
4 Oak Ridge National Lab, Oak Ridge, TN 37831, USA
* Correspondence: ahu3@utk.edu



**Featured Application: This paper can provide the basic knowledge on metal-air batteries for beginners and relevant comprehensive review for researchers.**

**Abstract:** With the ever-increasing demand for power sources of high energy density and stability for emergent electrical vehicles and portable electronic devices, rechargeable batteries (such as lithium-ion batteries, fuel batteries, and metal–air batteries) have attracted extensive interests. Among the emerging battery technologies, metal–air batteries (MABs) are under intense research and development focus due to their high theoretical energy density and high level of safety. Although significant progress has been achieved in improving battery performance in the past decade, there are still numerous technical challenges to overcome for commercialization. Herein, this mini-review summarizes major issues vital to MABs, including progress on packaging and crucial manufacturing technologies for cathode, anode, and electrolyte. Future trends and prospects of advanced MABs by additive manufacturing and nanoengineering are also discussed.

**Keywords:** metal–air batteries; laser processing; 3D printing

## 1. Introduction

### 1.1. Market Demand and Technical Tendencies

With the continued growth of the global economy, the demand for energy has significantly increased. Unfortunately, Earth's conventional non-renewable energy resources, such as coal, oil, and natural gas, are limited. Hence, the development of new energy devices is important for a sustainable society. Innovative biofuel batteries, supercapacitors, and metal–air batteries are among the most suitable candidates to meet the energy storage demand [1–6]. Among the various power storage devices currently on the market, lithium-ion batteries (LIBs) have the best performance. However, it is still a challenge to achieve high capacity (>200 mA h g$^{-1}$) in LIBs and to meet safe energy storage requirements for electric vehicles [7,8]. Recently, MABs attracted significant attention as they can operate in an open-air atmosphere. MABs consist of metal anodes and an air cathode. The MAB cathode uses oxygen from ambient air, which leads to significant battery weight reduction, which has unprecedented advantages for many applications. Compared to other batteries, especially Lithium-ion batteries, which currently dominate the market share, MABs are cheap, because the cathode source (oxygen from air) is abundant and the anode can be made using low-cost metals, such as, Al, Zn, Fe. Figure 1 shows the application of MABs as the energy storage system for various technologies. MABs

are attractive not only as compact power sources for portable electronics and electric vehicles but also as compelling energy transfer stations or energy storage devices to manage energy flow among renewable energy generators, such as wind turbines and photovoltaic panels, electric grids and end-users.

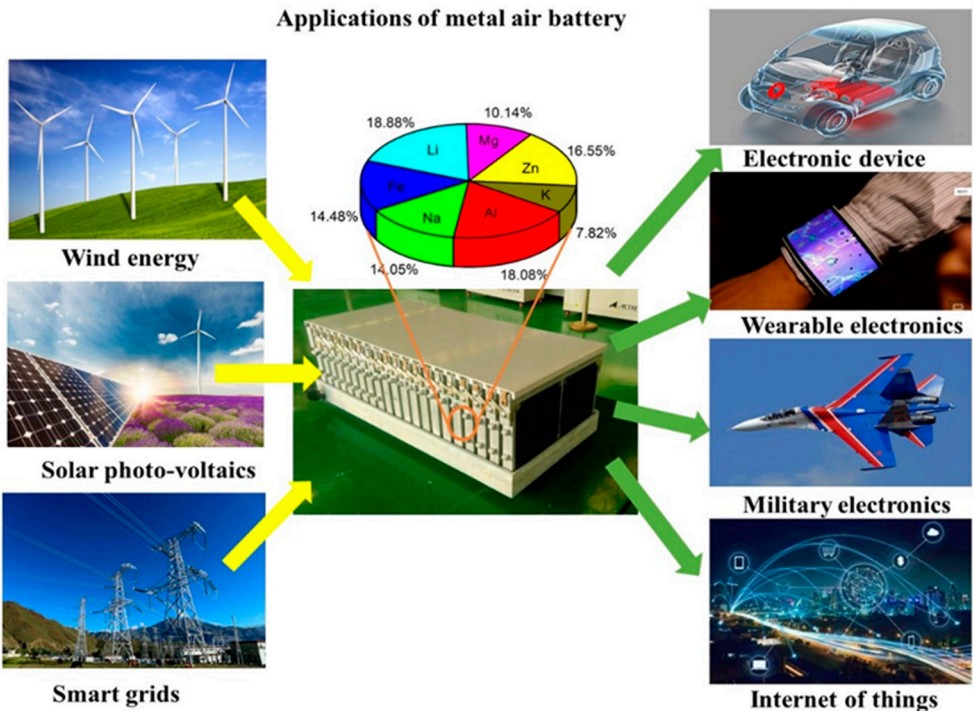

**Figure 1.** Applications of metal–air batteries as energy source and storage systems.

Theoretical energy density is an important factor in evaluating the performance of various battery configurations. Figure 2 shows theoretical energy density, specific energy, and nominal cell voltage of different metal-air batteries (MABs) [9]. As oxygen, directly supplied from the surrounding environment, is involved in the cathode as an oxidant during the discharge period, MABs show considerably higher energy density. Although, theoretically, lithium–air batteries (LABs) offer the best combination of the highest theoretical energy density (5928 Wh kg$^{-1}$) and high cell potential (nominally 2.96 V), iron–air batteries (FABs) possess the smallest theoretical energy density and cell voltage (nominally 1.28 V). Al-, Zn-, and Fe–air batteries are also the research hotspots because of economic and safety considerations.

In the present paper, aluminum–air batteries (AABs), zinc–air batteries (ZABs), iron–air batteries (FABs), and lithium–air batteries (LABs) have been reviewed with a focus on working principle and device configuration, and performance progress. In addition, major technology barriers have been identified, and possible solutions discussed. Emerging advanced manufacturing methods, such as 3D printing and laser processing techniques, for the development a high-performance rechargeable MABs, have also been discussed.

### 1.2. Working Principles

The working principle of MABs differs from that of traditional ionic batteries. The traditional ionic batteries involve the transformation of metallic ions from the anode to the cathode. In MABs, metals or alloys transform to metallic ions at anode and oxygen transforms to hydroxide ions at the cathode. Figure 3 shows the operation of a MAB in aqueous or non-aqueous electrolyte medium. In an aqueous electrolyte system, oxygen diffuses into batteries through the gas diffusion layer and transforms into receiving electrons forming oxygen anions. In a non-aqueous electrolyte system, oxygen receives electrons and transforms into oxygen anion. Metals release electrons, transform to

metallic ions and dissolve into electrolytes. These processes will be reversible during a charging procedure of a rechargeable MAB.

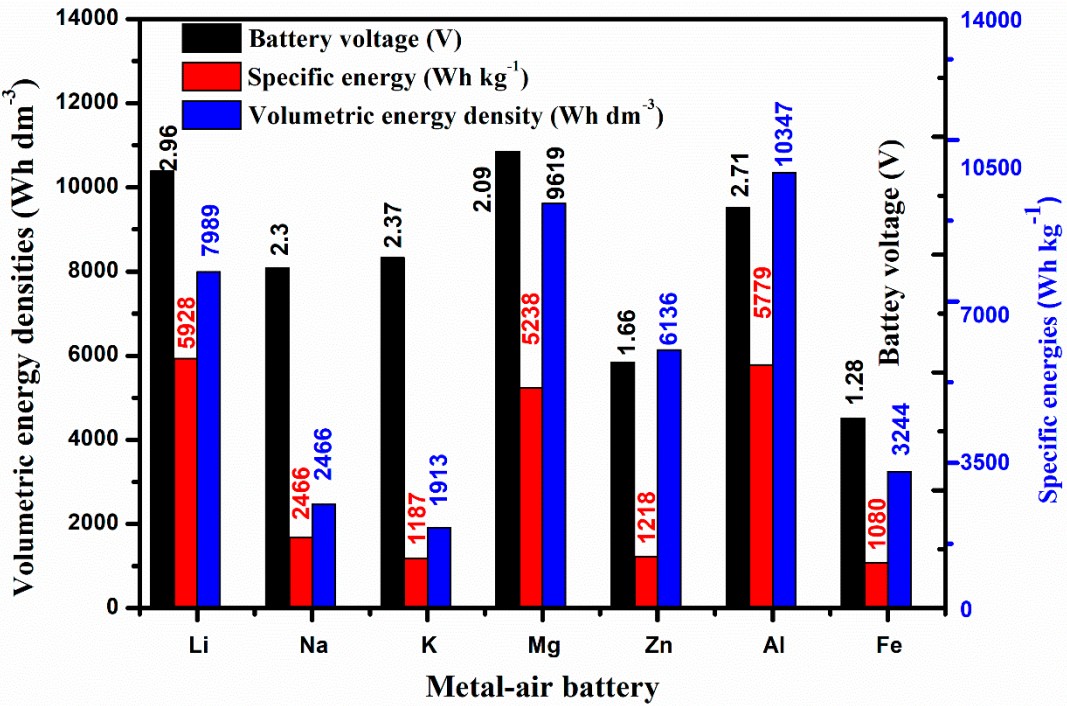

**Figure 2.** Theoretical specific energies, volumetric energy densities, and nominal battery voltages of various metal–air batteries (MABs) [9].

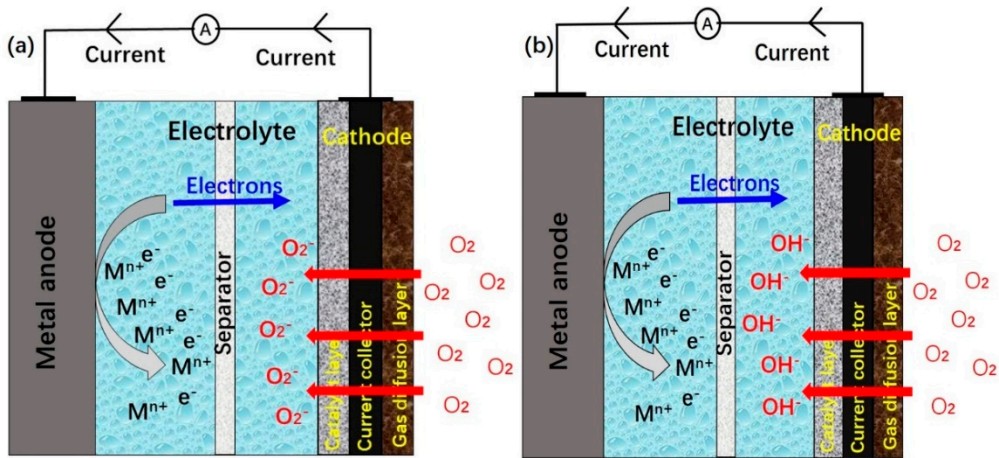

**Figure 3.** Schematic diagrams of MABs working principles for (**a**) non-aqueous electrolyte, and (**b**) aqueous electrolyte.

For MABs, oxygen and metals participate in electrochemical reactions. Specific reaction formulas are as Equations (1) and (2):

$$\text{Anode: M} \rightleftharpoons \text{M}^{n+} + n\text{e}^- \tag{1}$$

$$\text{Cathode: O}_2 + 2\text{H}_2\text{O} + 4\text{e}^- \rightleftharpoons 4\text{OH}^- \tag{2}$$

The reaction kinetics of FABs in the alkaline aqueous electrolyte are shown in Equations (1) and (6) [10].

$$\text{Anode: Fe} + 2\text{OH}^- \rightleftharpoons \text{Fe(OH)}_2 + 2\text{e}^- \tag{3}$$

$$3\text{Fe(OH)}_2 + 2\text{OH}^- \rightleftharpoons \text{Fe}_3\text{O}_4 + 4\text{H}_2\text{O} + 2\text{e}^- \tag{4}$$

$$\text{Cathode: } O_2 + 2H_2O + 4e^- \rightleftharpoons 4OH^- \tag{5}$$

$$\text{Overall reaction: } 2O_2 + 3Fe \rightleftharpoons Fe_3O_4 \tag{6}$$

The working principle of AABs in the alkaline aqueous electrolyte is shown in (9) [11].

$$\text{Anode: } Al + 4OH^- \rightleftharpoons Al(OH)_4^- + 3e^- \tag{7}$$

$$\text{Cathode: } O_2 + 2H_2O + 4e^- \rightleftharpoons 4OH^- \tag{8}$$

$$\text{Overall reaction: } 3O_2 + 2Al \rightleftharpoons Al_2O_3 \tag{9}$$

The working principle of ZABs in the alkaline aqueous electrolyte is shown in (13) [12].

$$\text{Anode: } Zn + 4OH^- \rightleftharpoons Zn(OH)_4^{2-} + 2e \tag{10}$$

$$Zn(OH)_4^{2-} \rightleftharpoons ZnO + 2OH^- + H_2O \tag{11}$$

$$\text{Cathode: } O_2 + 2H_2O + 4e^- \rightleftharpoons 4OH^- \tag{12}$$

$$\text{Overall reaction: } O_2 + Zn \rightleftharpoons ZnO \tag{13}$$

The working principle of LABs in the non-aqueous electrolyte is shown in (18) [1].

$$\text{Anode: } Li \rightleftharpoons Li^+ + e^- \tag{14}$$

$$\text{Cathode: } O_2 + e^- \rightleftharpoons O_2^- \tag{15}$$

$$O_2^- + Li^+ \rightleftharpoons LiO_2 \tag{16}$$

$$LiO_2 + Li^+ + e^- \rightleftharpoons Li_2O_2 \tag{17}$$

$$\text{Overall reaction: } O_2 + Li \rightleftharpoons Li_2O_2 \tag{18}$$

### 1.3. Configuration of MABs

Based on packaging and practical application requirements, MABs can be classified as traditional static batteries, flow batteries, and novel flexible batteries [13]. In this part, three kinds of batteries will be discussed briefly. The latter part is based on the analysis of solid-state batteries.

**Traditional static batteries:** As shown in Figure 3, traditional static air batteries have four main parts: cathode, separator, electrolyte, and anode. Compared to the fast kinetics of the anode reaction, oxygen reaction on cathodes is kinetically sluggish in nature. A three-phase reaction boundary of solid (catalyst)–liquid (electrolyte)–gas (oxygen) contributes to the oxygen reduction reaction (ORR). Meanwhile, a reversed oxygen evolution reaction (OER) occurs on a two-phase boundary solid (catalyst)–liquid (electrolyte) at the cathode [14]. Highly efficient bifunctional catalysts are thus required to facilitate both OER and ORR. In addition, since an active electrolyte is employed in traditional static MABs, it is challenging to completely overcome the issue of insoluble deposition of by-products on the surface of both the metal anode and air cathode during the charge–discharge cycles. These deposited by-products consequently block the electrode pores limiting the diffusion of air that eventually results in a lower battery performance [15].

**Flow batteries:** This type of MAB consists of an electrode, separator, electrolyte, and an electrolyte bank installed as an additional part. Usually, a pump is also integrated to drive electrolyte flow, as shown in Figure 4b. Flowing-electrolyte configuration addresses some of the problems associated with the metal anode and an air cathode. For instance, in zinc–nickel batteries, a large volume of flowing electrolyte decreases the formation of dendrites and irregular shape changes of zinc and thereby, avoid passivation by improving current distribution and reducing concentration gradients [16]. However, the complicated flowing-configuration of MABs has some shortcomings, including decreased

energy efficiency and volume density, and increased complications as additional pumps and tubes are needed to drive the flow of electrolyte during the discharge.

**Flexible batteries**: With the increasing demand for portable electronics in recent years, light and small form-factor flexible batteries have become a hot research topic [17,18]. The main components of a flexible battery include a cathode, anode, separator, and high conductivity electrolyte. Conventional electrolyte for a flexible battery system is a solid-state electrolyte. A thin metallic plate is used as a metal anode to reduce the battery weight. Various nano compounds and nanocomposites are being explored as a potential cathode material. Specific materials include carbon fibers, carbon nanotubes, and graphene. According to the current development trend, flexible batteries are undergoing an evolution, from polymer batteries, flexible alkaline batteries, lithium-based batteries to metal–air batteries. Currently, ZABs and AABs are the ideal flexible batteries due to low cost, safe operation, and high energy density [19].

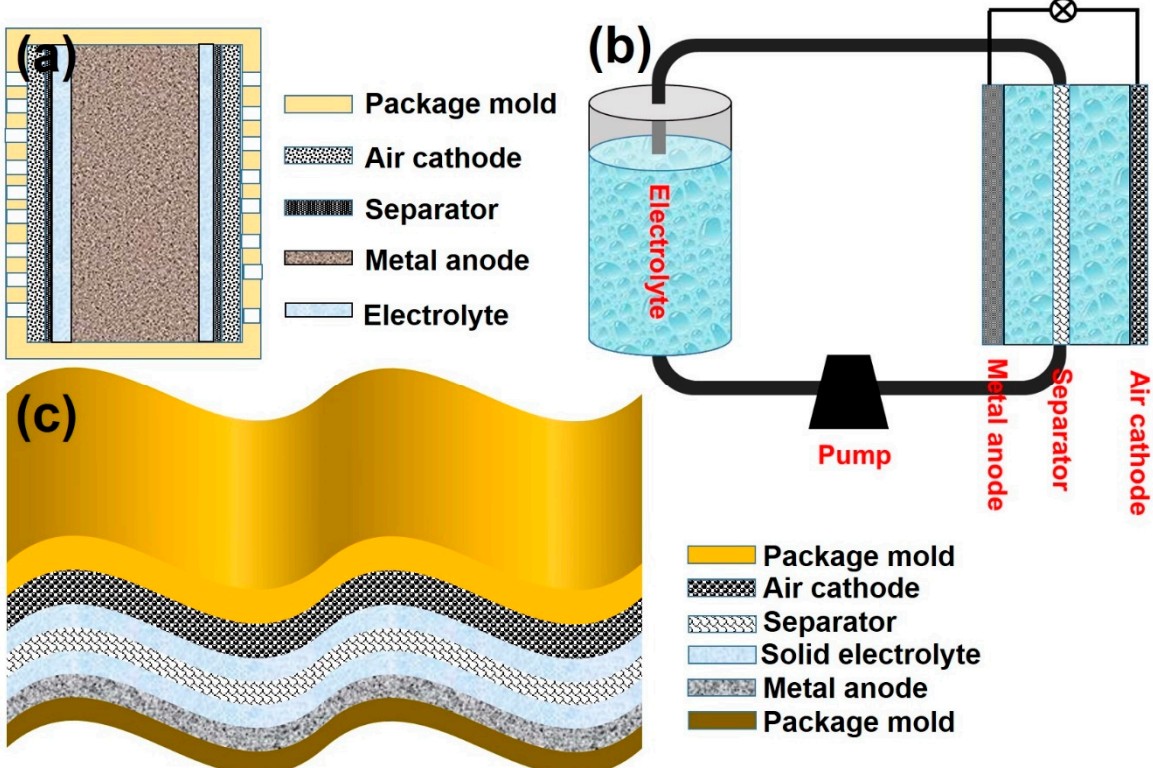

**Figure 4.** Schematic diagram of different MAB configurations: (**a**) multi-cell static configuration, (**b**) flow battery, and (**c**) flexible battery.

## 1.4. Technical barriers

Although metal–air batteries have been studied for many years, there are still major technical issues to address for practical applications. Metallic anodes face many challenges, such as corrosion, hydrogen generation, forming passivation layers, dendritic formation, electrode deformation, and energy loss due to self-charging. The air anode has many obstacles, such as lack of efficient catalysts for both ORR and OER, affecting electrolyte stability due to impurity and dissolved gas, and gas diffusion blockage by side reaction products. Electrolyte selection, which is an important component for efficient electrochemical reaction, also poses some technical barriers due to side reaction with the anode, reaction with $CO_2$ from air, and low conductivity. In the following sections, we will discuss these issues and potential solutions.

## 2. Cathodes

### 2.1. Components of the Cathodic Electrode

On the cathode, chemistry reactions are ORR and OER. The oxidant is oxygen from air atmosphere. The catalysts are required to lower overpotential of ORR and OER. For an aqueous electrolyte, water loss should be avoided to keep battery stability. Hence, the practical cathode is composed of a catalyst layer, gas diffusion layer, and current collector, as shown in Figure 5. The current collector can be metal and non-metal. Metal current collectors are a porous foam-like metal, for example, Ni, Cu. Non-metal current collectors are carbon-based material, for example, conductive carbon paper, graphitic fiber or carbon cloth. Gas diffusion layer (GDL) and the catalysts are also extremely crucial for cathode performance.

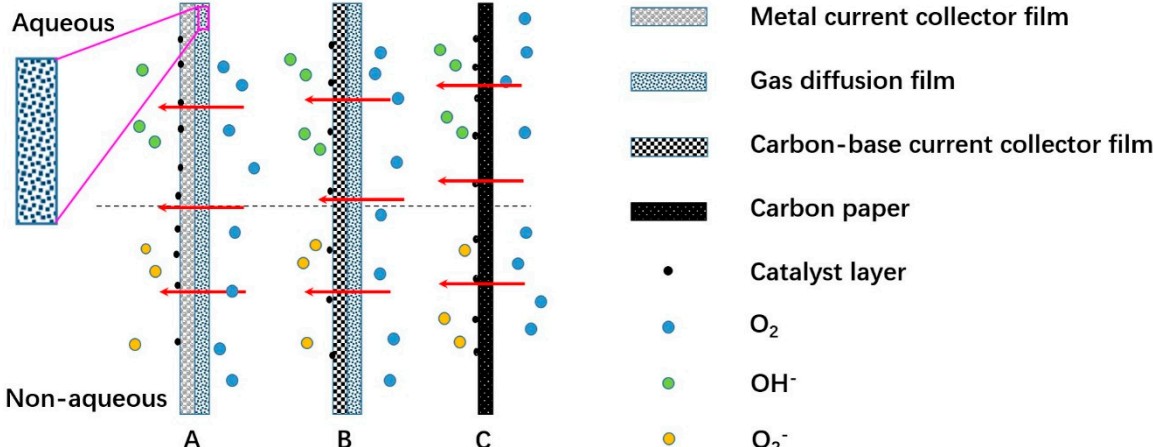

**Figure 5.** Various cathodic electrode of MABs, (**A**) metal current collector with a gas diffusion layer and coated catalyst facing electrolyte, (**B**) carbon-base current collector and gas diffusion and catalyst, (**C**) carbon paper-current collector and gas diffusion and catalyst.

(1) **Gas diffusion layer**: In MABs, the GDL has multi-folded functions: supporting of catalyst layer; providing oxygen diffusion channels between air and catalyst layer; preventing water getting into battery and electrolyte getting out of battery. To better serve as a bridge between air and catalyst layer, the GDL should be thin, light, highly porous, and hydrophobic. Figure 4b indicates that the ORR in MABs occurs at the three phase boundaries (oxygen air, liquid electrolyte, and solid catalyst). The GDL can simultaneously provide hydrophilic micro-channels to the liquid electrolyte, and hydrophobic layers to prevent electrolyte leakage and good properties of gaseous oxygen diffusion [20].

(2) **Catalyst layer**: Since the kinetics for the oxygen reaction is naturally slow, bifunctional catalysis is required to improve ORR and OER to improve electrochemical performances of MABs. Based on previous researches, platinum (Pt) [21], ruthenium (Ru) oxides, and iridium oxides (Ir) [22] showed excellent performance in ORR and OER. Furthermore, nanostructures of the following materials also had good catalytic activity, (a) transition metal oxides, MnO, CoO, NiO, etc. [23]; (b) transition metal hydroxide and sulfide, NiCoFe-LDH (Layered double hydroxides) layered double hydroxides [24]; (c) spinel compounds, such as $CuCo_2O_4$ [25]; (d) carbon-based materials, such as nitrogen doping carbon [26]; (e) nanocomposite materials mixing ORR catalyst Fe-N-C and OER catalyst NiFe [27].

### 2.2. Improving ORR and OER

The appropriate catalyst should be designed and applied to maximize catalytic efficiency. Metal-based catalysts possess high catalytic efficiencies due to different crystal structures. Spinel-type oxide ($A_xB_{3-x}O_4$) [28] and perovskite oxides ($ABO_3$) [29] are widely used as bifunctional electrocatalysts in alkaline electrolytes. Maiyalagan et al. [30] synthesized a spinel-type lithium cobalt oxide $LiCoO_2$ at

high-temperature (800 °C, LiCoO$_2$-HT) and low-temperature LiCoO$_2$-LT at 400 °C. LiCoO$_2$-LT adopts a lithiated spinel structure [Li$_2$]$_{16c}$[Co$_2$]$_{16d}$O$_4$ in which the Co$^{3+}$ ions occupy all the 16d octahedral sites (space group: Fd3m) [31]. As shown in Figure 6a, LiCoO$_2$-HT has the α-NaFeO$_2$ structure (space group: R3m) arrays. Li$^+$ ions and Co$^{3+}$ ion occupy on alternate (111) NaFeO$_2$-style structure arrays, due to the large size and charge differences between the Li$^+$ and Co$^{3+}$ ions [32]. In Figure 6b, it is obvious that HT-LCoO$_2$ has a better catalytic performance than LT-LiCoO$_2$ and Co$_3$O$_4$.

Catalysts are preferred at nanoscale for better catalytic behavior. By using a vacuum DC arc method, Lang et al. [33] synthesized a novel Mn$_3$O$_4$/MnO nano spherical transition metal compound. The results showed that the size of Mn$_3$O$_4$/MnO particles was controlled at a range of 40 to 60 nm. The Mn$_3$O$_4$/MnO catalyst potential platform reaches to 2.7 V.

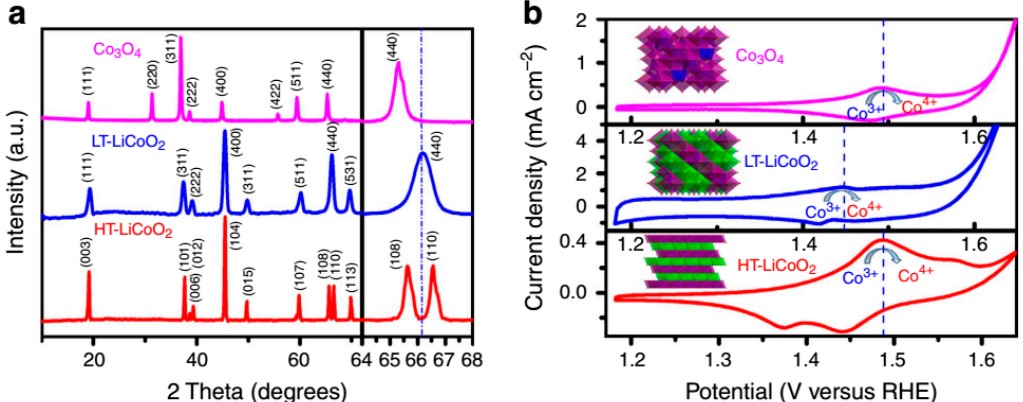

**Figure 6.** (**a**) X-ray diffraction patterns and (**b**) electrochemical behaviors of Co$_3$O$_4$, low-temperature (LT)-LiCoO$_2$, high-temperature (HT)-LiCoO$_2$ catalysts [30] (Copyright © 2014, Springer Nature).

Nanoscale catalysts can also be fabricated with various other morphologies, such as nano-rod LaCoO$_3$ [34], 3D ordered mesoporous structure Co$_3$O$_4$ [35], hollow cobalt oxide nanoparticles [36]. Different morphologies of catalysts are shown in Figure 7. Many low-cost and efficient catalysts have been developed, including transition metals and nitrogen co-doped carbons (M-N/C, M=Fe or Co) [25], metal oxides [37], transition metal carbides [38], nitrides [39], and metal-free heteroatom-doped carbon-based catalysts [40]. Compared to metal-contained catalysts, the heteroatom-doped carbon-based materials with N, S, B, and P, can promote oxygen adsorption on the carbon nanostructure since these hetero-atoms are more electronegative than carbon, and cause neighboring carbon atoms electron deficiency [41]. Among them, N-doped carbons are extensively studied due to their remarkable ORR catalytic activity. N-doped carbon materials are shown in four ways, graphitic N, Oxidized N, pyrrolic N, and pyridic N in Figure 8.

Although several materials have shown catalytic activity for oxygen reaction in MABs cathodes, the catalytic efficiency is not ideal when used alone. To improve comprehensive catalyst performance, composition materials have been synthesized and used as catalysts in both ORR and OER. MnO$_2$ and RuO$_2$ are single catalysts for ORR and OER, respectively. Combining MnO$_2$ with RuO$_2$ is used as a bifunctional catalyst. Sun et al. [42] synthesized RuO$_2$ nanoparticles (np-RuO$_2$/nr-MnO$_2$) supported on MnO$_2$ nanorods by a two-step hydrothermal reaction. Electrochemical characterizations are carried on nanocomposites np-RuO$_2$/nr-MnO$_2$ as catalysts for LABs. Charge–discharge tests showed a reversible discharge capacity of 500 mAh g$^{-1}$ for 75 cycles at a current density of 50 mA g$^{-1}$. LABs with the RuO$_2$/MnO$_2$ catalyst presented much lower overpotential of 0.58 V at 50 mA g$^{-1}$ than that measured with a single catalyst. ORR and OER electrocatalytic activity were tested by using rotating disk electrodes. It was found that np-RuO$_2$/nr-MnO$_2$ ORR limitation diffusion current was 6.01 mA cm$^{-2}$, and ORR half-wave potential (E$_{1/2}$) was −0.158 V. These results demonstrated that an np-RuO$_2$ and nr-MnO$_2$ combination can work as an effective catalyst for LABs with high activity while maintaining batteries stability.

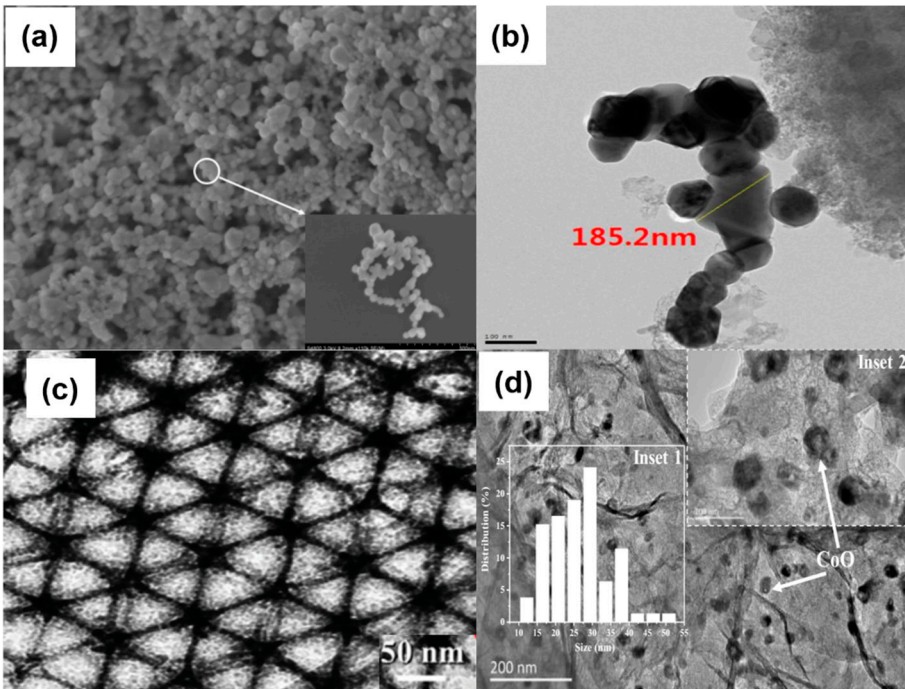

**Figure 7.** (**a**) SEM images of Mn₃O₄/MnO nanoparticles [33] (Copyright © 2019, Elsevier), (**b**) TEM images of LaCoO₃ catalysts [34], (**c**) TEM image of honeycomb-like 3D ordered mesoporous spinel Co₃O₄ [35] (Copyright © 2016, John Wiley and Sons), (**d**) TEM image of hollow cobalt oxide nanoparticles [36] (Copyright © 2019, Elsevier).

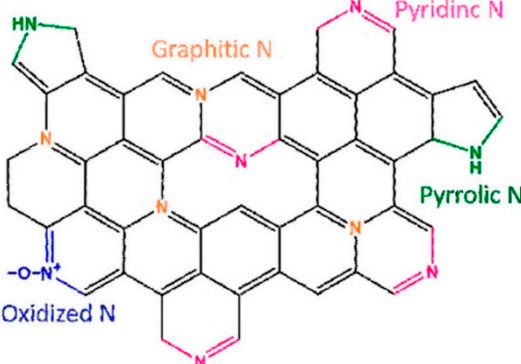

**Figure 8.** Four nitrogen doping configurations of a graphene molecule [23].

### 2.3. In Situ Characterization Using an Electron Microscope

An in situ electron microscope is a promising tool for scientific research due to real-time observation and plays a major role in many MAB studies, such as the catalytic mechanism, the oxidation–reduction mechanism, the growth of nanostructures, and the deformation of electrodes. Based on the distinctive features of the preceding, Katharine et al. [43] found that higher Coulombic efficiency and more homogeneous morphology of the Li deposits in a coin-cell contributed to the presence of a compressed lithium separator interface through in situ electrochemical-scanning transmission microscope (EC-STEM), compared with a macroscale cell. In addition, Yoon et al. [44] used in situ atomic force microscope (AFM) to measure the dominant wavelength of the wrinkled surface topography. The planes strain modulus of the SEI was determined from the measured wavelength. Li et al. [45] explored the reaction mechanism and unveiled that $\alpha$-MoO₃ converted to crystalline Li₂MoO₃ in the first stage of lithiation, and further converted to metallic Mo and amorphous Li₂O in the next stage. As shown in Figure 9a, with a negative potential to the Au/MnO₂ nanowires (NWs),

bubble-like $NaO_2$ nucleated on the contact where the $Au/MnO_2$ NW and $Na_2O$ intersects, then grows along the NW, resulting in 18 times volume increase. Meanwhile, the discharge product shrinks as a result of the disproportionation of $NaO_2$ to $Na_2O_2$ and $O_2$, confirming the occurrence of ORR [46]. Similarly, Liu et al. [47] also reported the real-time observation of ORR in Figure 9b. In this research, CuO nanowires (NWs), as the air cathode, firstly converted to $Cu_2O$ and then to Cu, as a metal catalyst to accelerate the disproportionation of $NaO_2$ to $Na_2O_2$ and $O_2$. During the reaction process, the morphological changes were investigated by an electrochemical atomic force microscope. Liu et al. [48] used EC-AFM to observe the dynamic process of $Li_2O_2$ growth/decomposition during the ORR/OER on a gold electrode in Figure 9c and found that the $Li_2O_2$ decomposed at a lower potential due to electrochemically generated TTF$^+$ through a homogeneous oxidation mechanism.

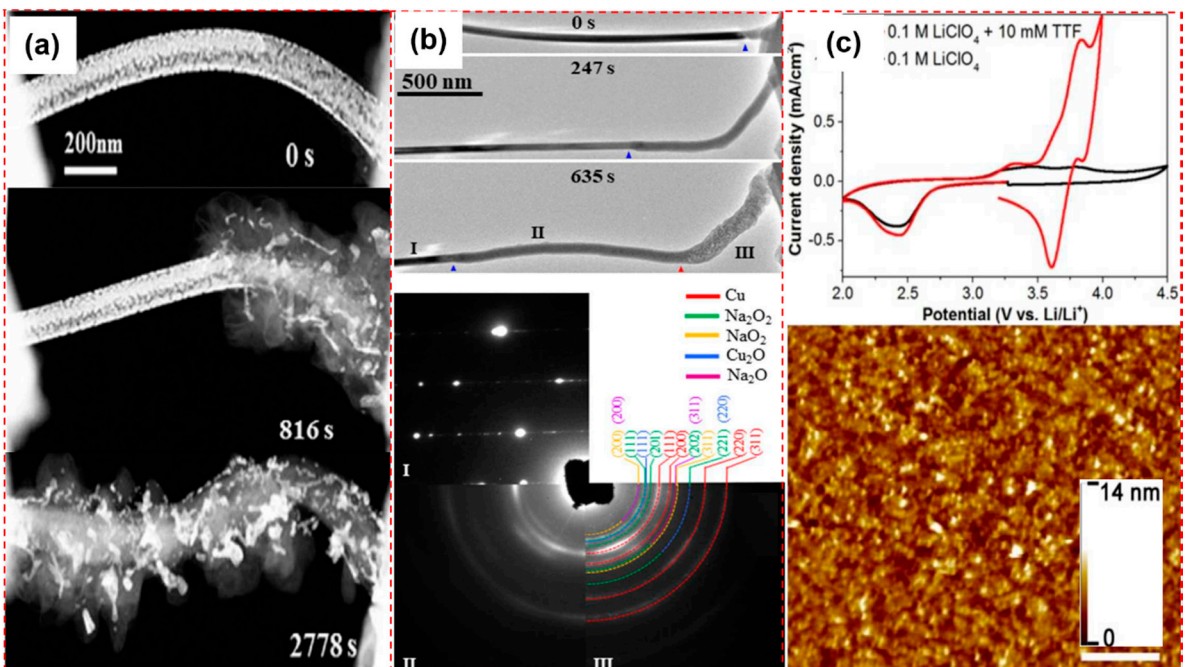

**Figure 9.** (**a**) Structure evolution of the $NaO_2$ discharge product during oxygen reduction reaction (ORR) [46] (Copyright © 2019, Elsevier); (**b**) Structural and phase characterization of a CuO nanowire (NW) during discharging and charging in an $O_2$ environment [47] (Copyright © 2018, American Chemical Society); (**c**) Cyclic voltammetry performed in electrochemical-atomic force microscope cell and the resulting AFM image after CV reduction [48] (Copyright © 2016, American Chemical Society).

## 3. Anodes

The chemical activity of the metal anode determines the discharge capacity. Because of high metal activity, an unavoidable side reaction with various components in the electrolyte may occur. Depending on the purity of the metal, the battery performance and the incidence of side reactions can be different.

### 3.1. Anode Materials: High Purity Metal and Alloy

Fan et al. [49] took industrial 5 N Al (99.999% high purity) and aluminum alloy (1050, 2011, 3003, 4032, 5052, 6061, 7050, and 8011) as anodes for AABs in alkaline electrolytes, using the hydrogen collection method and electrochemical impedance spectroscopy (EIS) to determine the corrosion behaviors, electrochemical properties, and potentiodynamic polarization. Test results of corrosion and EIS showed the sample in 4 M KOH was more suitable than in 4 M NaOH. Al 8011 had a transfer resistance ($R_t$) of 1.247 $\Omega$ cm$^2$ in 4 M NaOH and 1.108 $\Omega$ cm$^2$ in 4 M KOH. 5 N Al had $R_t$ of 2.29 $\Omega$ cm$^2$ in 4 M NaOH and 15.3 $\Omega$ cm$^2$ in 4 M KOH. 5 N Al had 1.699 V $E_{corr}$ in 4M NaOH and 1.821 V $E_{corr}$ in 4M KOH. All industrial Al alloy anodes had the hydrogen adsorption phenomenon of hydrogen

evolution reaction, and 8011 had relatively better performance than the others. As shown in Table 1, Al 8011 had a lower corrosion potential ($E_{corr}$) at −1.42 V, corrosion current ($I_{corr}$), of 135 mA cm$^{-2}$, and polarization resistance ($R_p$) of 3.628 Ωcm$^2$ among industrial Al alloys. Therefore, impurity elements had different roles in the corrosion behaviors. Mg, Mn, Cr, Ti, and Zn were helpful to improve the corrosion resistance of Al alloy anodes, while Fe, Cu, and Si formed cathodic sites and lowered overpotential for hydrogen evolution reaction (HER) [49–51]. Different components Al alloys (Zn-rich and Al-rich phases) worked as abode AABs anode. Test results showed Al-rich alloys were better performance, due to lower the anodic passivation. Zn-Al alloys are promising anode materials as primary and mechanical-rechargeable Zn-air batteries [52].

**Table 1.** Parameters and electrochemical impedance spectroscopy (EIS) value of different grades of Al anodes [49].

| Grade | $E_{corr}$ (V vs. Hg/HgO) | | $I_{corr}$ (mA cm$^{-2}$) | | $R_p$ (Ω cm$^2$) | |
|---|---|---|---|---|---|---|
| | 4 M NaOH | 4 M KOH | 4 M NaOH | 4 M KOH | 4 M NaOH | 4 M KOH |
| 1050 | 1.290 | 1.291 | 186 | 176 | 4.108 | 5.683 |
| 2011 | 1.410 | 1.420 | 135 | 142 | 4.124 | 6.674 |
| 3003 | 1.315 | 1.340 | 181 | 165 | 4.607 | 5.892 |
| 4032 | 1.390 | 1.390 | 145 | 174 | 4.653 | 2.572 |
| 5052 | 1.310 | 1.320 | 191 | 157 | 3.766 | 2.539 |
| 6061 | 1.370 | 1.380 | 161 | 168 | 4.766 | 2.655 |
| 7050 | 1.420 | 1.450 | 189 | 143 | 4.516 | 2.848 |
| 8011 | 1.420 | 1.450 | 135 | 144 | 3.628 | 2.670 |
| 5N | 1.699 | 1.821 | 24.3 | 4.7 | 9.668 | 17.9 |

| Equivalent elements | | Grade | | | | | | | | |
|---|---|---|---|---|---|---|---|---|---|---|
| | solution | 1050 | 2011 | 3003 | 4032 | 5052 | 6061 | 7050 | 8011 | 5N |
| $R_t$ (Ω cm$^2$) | 4 M NaOH | 0.70 | 0.75 | 0.57 | 0.84 | 0.74 | 0.49 | 0.76 | 1.247 | 2.29 |
| | 4 M KOH | 1.23 | 0.69 | 1.208 | 0.3351 | 0.78 | 0.36 | 0.60 | 1.108 | 15.3 |

## 3.2. Metal Coating and Composite Electrodes

Different from previous research using Al alloy as anodes, Mutlu [53] investigated Cu coating on Al and 7075 Al alloy as anodes. Copper was deposited on the Al surface by chemical (Al or Alloy/Cu-CD) and electrochemical (Al or Alloy/Cu-ED) processes, SEM images are shown in the Figure 10a,b. Al has a lower resistance than Al-Cu alloy. EIS measurements showed that copper on the Al surface could decrease anodic potential and improve batteries performance as shown in Figure 10c,d. The solution–electrode interaction resistance ($Rs$) was increased by adding copper to aluminum because copper can form a protective layer against the corrosion on the aluminum–solution interface. Hang et al. [54] found that FeS can employ as an additive for the electrode to suppress hydrogen evolution and improve the cyclic performance of the Fe/C composite anode. FeS additive and the carbon component also strongly affected the redox behavior of iron. An electrode with FeS can promote the process (in the Figure 11) of $Fe^0$ to $Fe^{2++}$ and $Fe^{2+}$ to $Fe^{3+}$ at around −0.85 V ($a_1$), −0.65 V ($a_2$), respectively. Furthermore, there were also additional intermediate species which appeared around −0.97 V ($a_0$). The discharge capacity of the electrode was significantly improved with adding 2 wt.% FeS, which was mainly because the incorporation of FeS in the electrode improved the adsorbed capability of $S^{2-}$ on the electrode surface, resulting in the easy breaking of the oxide layer. For HER at the anode in alkaline electrolyte, both molecular recombination and electrochemical desorption can be parallel steps in the overall process. The molecular recombination reaction will appreciably contribute to HER only when the current density is low, and the molecular hydrogen concentration at the liquid boundary layer is near to zero. Under these conditions, it has been indicated that the molecular recombination reaction is affected by $S^{2-}$ ion chemisorption.

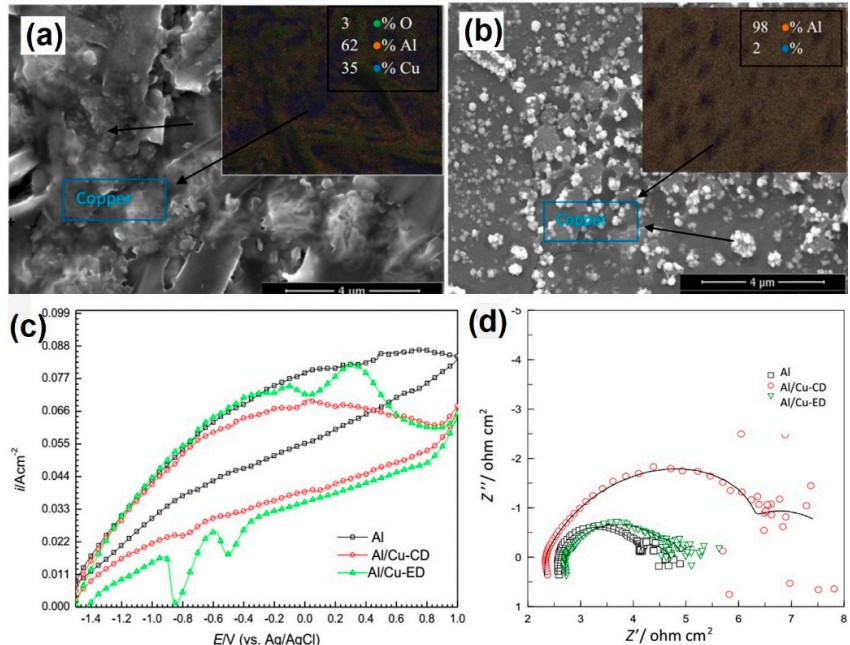

**Figure 10.** SEM images and EDS maps of (**a**) Al/Cu-CD, (**b**) Al/Cu-ED, (**c**) The cyclic voltammogram in 1 M NaOH of Al (pure), Al/Cu-CD, and Al/Cu-ED, (**d**) EIS measurements of anodes in 1 M NaOH, The Nyquist a Al (pure), Al/Cu-CD, Al/Cu-ED [53] (Copyright © 2018, Springer Nature).

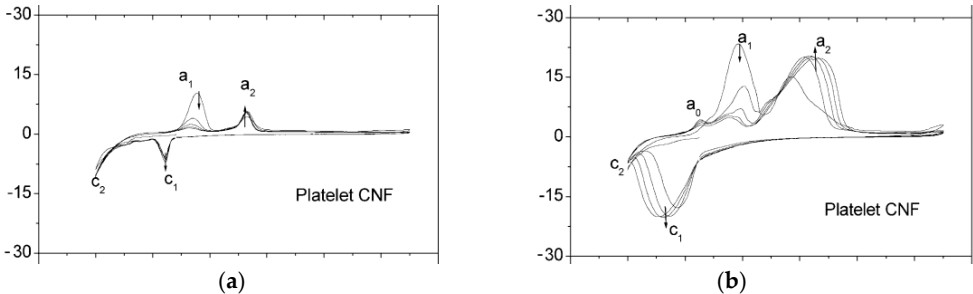

**Figure 11.** Cyclic voltammetry for Fe/C composite electrodes composited carbon nano-fibers (**a**) without and (**b**) with FeS additive [54] (Copyright © 2006, Elsevier).

### 3.3. Common Challenge of Metal Anode

The common issues with metallic anodes are corrosion, passivation, and dendrite formation. These mechanisms are displayed in Figure 12.

Corrosion: Corrosion is one of the major side reactions between metal and electrolyte, and its reaction can be expressed as follows:

$$M + (2 + x)H_2O \rightleftharpoons 2M(OH) + H_2 \tag{19}$$

$$M + H_2O \rightleftharpoons MO_X + H_2 \tag{20}$$

Equations (19) and (20) evaluates the corrosion rate due to hydrogen evolution reaction (HER). For almost all MABs, the M/MO standard voltage was below that of the hydrogen revolution. Therefore, hydrogen evolution was spontaneously favored. The HER decreased metal anode Coulombic efficiency because it consumed electrons from the metal anode in the charge. Moreover, hydrogen diffusing into electrolyte leads to the increase of internal battery pressure and could result in an explosion.

**Hydrogen evolution reaction:** Hydrogen evolution reaction (HER) was a side reaction of metal electrodes during the charge–discharge of batteries. The specific working principle is shown in Equations (19) and (20). Metal releases electrons to the aqueous electrolyte system and hydrogen ions

replace metal ions obtaining electron reduction in hydrogen. HER in MABs thus influences the rates of metal electrodes. Hydrogen overpotential decreases on the ZnO surface since the self-discharge rate reduced with increasing ZnO on the electrode surface [55]. Increasing overpotential of HER (decreased HER rate) can thus improve the charging efficiency. In addition, the corrosion and oxidation of Al in alkaline electrolytes depend on electrolyte properties, temperature, and purity [56]. Using ionic liquid or solid state electrolyte is an effective solution to reduce the HER rate, which has been confirmed in FABs [57]. An alloy as anodes replacing pure metal also reduced the corrosion rate, and additives, such as bismuth or sulfur, could minimize the corrosion of the iron electrode and the evolution of hydrogen [58].

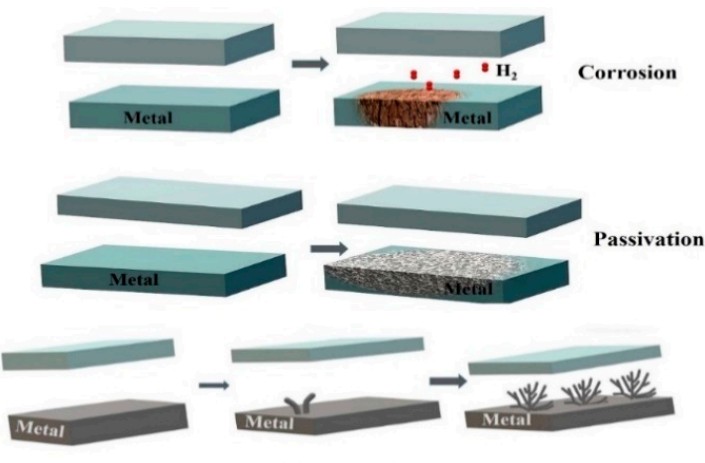

**Figure 12.** Corrosion, passivation, and dendrite formation processes at a metal anode.

**Passivation layers**: Passivation used to describe an electrode that could not be further discharged because an insulating film on its surface blocked migration of the discharge product. In MABs, LiOH, ZnO, and $Al_2O_3$ were passivation layers for corresponding systems. Soluble species formed at the air cathode will be reduced to a non-conductive layer on the metal surface. This a non-conductive layer increases the internal electrical resistance of the cell and prevents metal dissolution. The efficient method is to use porous electrodes to hinder the formation of passivation layers.

**Dendritic formation and deformation**: During the metal electrode cycling in an alkaline electrolyte, the metal anode releases ions during discharge and the metal ions re-deposit on the surface of the anode during charging. As a result, the metal electrode will gradually change shape, and its surface will become roughened with uneven thicknesses. Over several charge and discharge cycles, the uneven shape accumulates to form dendrites, causing the battery system to become unstable or short cut. Different approaches have been attempted to mitigate dendritic formation and deformation, such as coating the zinc metal and using non-reaction additives in the zinc electrode or electrolyte [59]. Lithium alloying with Na [60], Mg [61], Al [62] has been confirmed to effectively suppress the growth of dendritic Li.

In summary, passivation layers happen in AABs, ZABs, and LABs, dendritic structures form in ZABs, and LABs and FABs, and corrosion may occur in FABs, AABs, ZABs, and LABs. Different strategies are needed to address these issues.

## 4. Electrolytes

An electrolyte is a medium to transport ions and electrons to ensure the continued oxidation–reduction reaction. Electrolyte divided into four types: aqueous, non-aqueous, hybrid, and solid state, as shown in Figure 13.

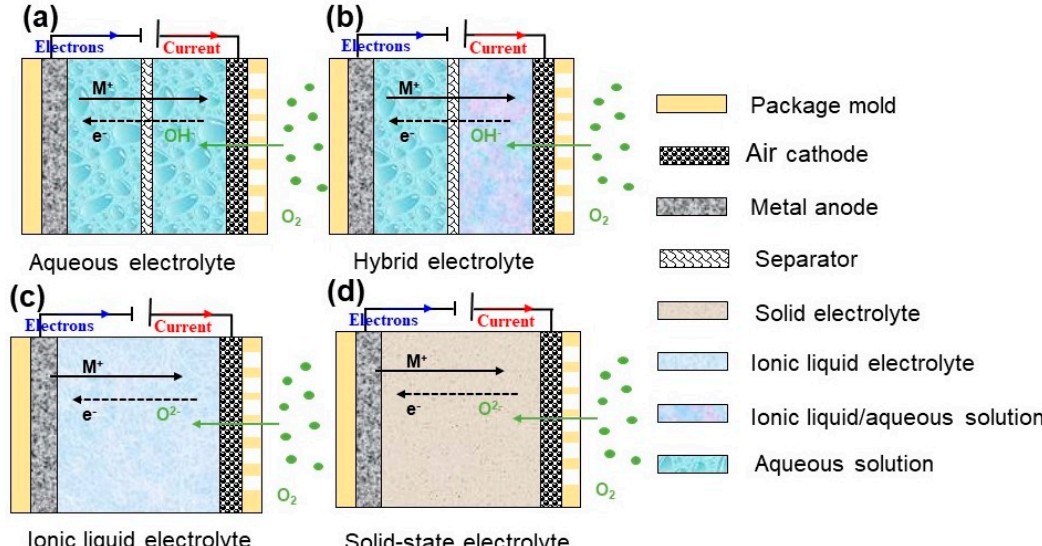

**Figure 13.** Schematic diagrams of various electrolytes of MABs.

## 4.1. Aqueous Electrolyte

**Alkaline solutions** ($7 < pH \leq 14$): Alkaline electrolytes are the most applied electrolyte in aqueous-based MABs, because the ORR is more favorable with faster reaction kinetics and a lower overpotential, compared to acidic electrolytes. Alkaline electrolytes have a shortcoming that $CO_2$ (from air atmosphere) reacts with electrolyte and forms a carbonate surrounding the cathode. The large amount of carbonate will block the porous structure of the positive electrode material and decrease cathode efficiency.

**Neutral salt solution** ($pH = 7$): Al alloy–air batteries can discharge in a neutral salt solution with a lower corrosion rate and higher activity than in an alkaline electrolyte.

**Acidic solutions** ($2 \leq pH < 7$): Acidic electrolytes are rarely used in aqueous-based MABs because a large amount of $H^+$ in solution could directly react with metal and reducing battery efficiency. Saidman et al. [63] reported Al-Zn alloy anode performance changed with various types of acids at the same concentration, pH, and operating temperature. Electrochemical test results indicted Al–Zn alloy in 0.5 M HCl was more negative than that in 0.5 M HAc: −1.02 V and −0.80 V, respectively.

**Hybrid electrolyte**: In Figure 14b,c, a novel type of aqueous FABs was equipped with an alkaline anode electrolyte (anolyte) and an acidic cathode electrolyte (catholyte). The anolyte and catholyte are separated by an alkali metallic ion ($Li^+$ or $Na^+$) solid-state electrolyte separator. The alkali metal ion serves as an ionic mediator to sustain the redox reactions at both the anode and cathode [64].

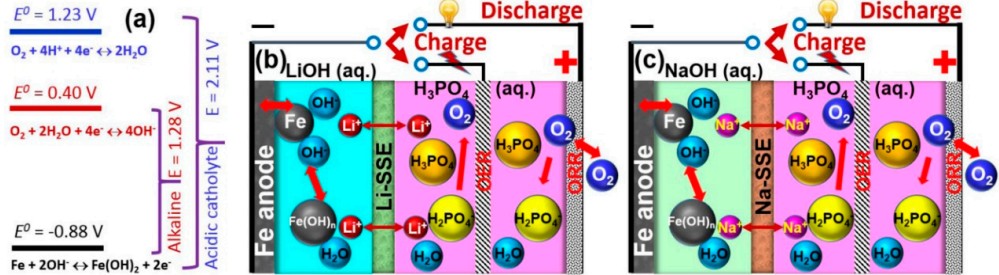

**Figure 14.** (**a**) Schematic illustration of theoretical voltages of Fe−air batteries operated with an alkaline or an acidic cathode electrolyte. Two types of Fe−air batteries with a $Na^+$- or a $Li^+$-ion solid electrolyte, for (**b**) a Fe(LiOH)//Li-SSE//$O_2$($H_3PO_4$/$LiH_2PO_4$) cell, and (**c**) a Fe(NaOH)//Na-SS//$O_2$($H_3PO_4$/$NaH_2PO_4$) cell. SSE represents solid-state electrolyte [64] (Copyright © 2017, American Chemical Society).

## 4.2. Non-Aqueous Electrolyte

**Solid-state electrolyte**: Solid-state electrolytes are different from aqueous electrolytes in dual characteristics of wettability and ion conduction. For MABs, aqueous electrolytes with an excellent wetting property at three-phase boundaries could be in full contact with the cathode. For a solid-based electrolyte, the three-phase interface reaction can be restricted by the poor wetting property of the "immobilized" electrolyte, thereby, interfacial transporting resistance of $OH^-$ may be remarkably higher than that of an aqueous system [65]. Alkaline gel electrolytes (AGEs), consisting of low molecular weight polymer and alkaline solutions have been developed to mitigate these issues for primary lithium–air batteries [66].

**Ionic liquid electrolyte**: Ionic liquids are non-aqueous liquid electrolytes, including two types of cations: large organic cations with organic/inorganic anions and alkali metal ions in an organic solvent, such as organic carbonates, ethers, and esters [67,68]. Lithium salt, such as $LiPF_6$, $LiAsF_6$, $LiN(SO_2CF_3)_2$, and $LiSO_3CF_3$ are commonly used in LABs [69]. PYR14TFSI–TEGDME–$LiCF_3SO_3$ are also employed in LABs [70], consisting of $LiCF_3SO_3$ in tetraethylene glycol dimethyl ether (TEGDME) and pure PYR14TFSI. Ionic liquid electrolytes also face challenges because of the formation of carbonates, which consume the electrolyte and block the electrode pores. What is more, the understanding of the oxygen reaction in an ionic liquid is very limited. These hinder the practical application of ionic liquid electrolytes.

In summary, neutral salt solution, acidic solutions, and hybrid electrolyte are rarely used in industrial application. Alkaline solutions electrolytes and ionic liquid electrolyte are usually employed. Aqueous electrolytes do not match LABs. Furthermore, solid-state electrolytes can work in all MABs.

## 5. Advanced Manufacturing of MABs

Advanced manufacturing techniques for electrodes and batteries is composed of printing and laser processing. Printing technology has various advantages in microstructure controlling and large batch low-cost fabrication. Printing includes screen printing [71], spray printing [72], direct ink writing processing [73], and roll-to-roll fabrication, as shown in Figure 15. Figure 16 shows printed 1- to 4D structures. 3D and 4D printing are achieved by layer-to-layer printing of functional micro/nanomaterials in geometric and temporal complexes.

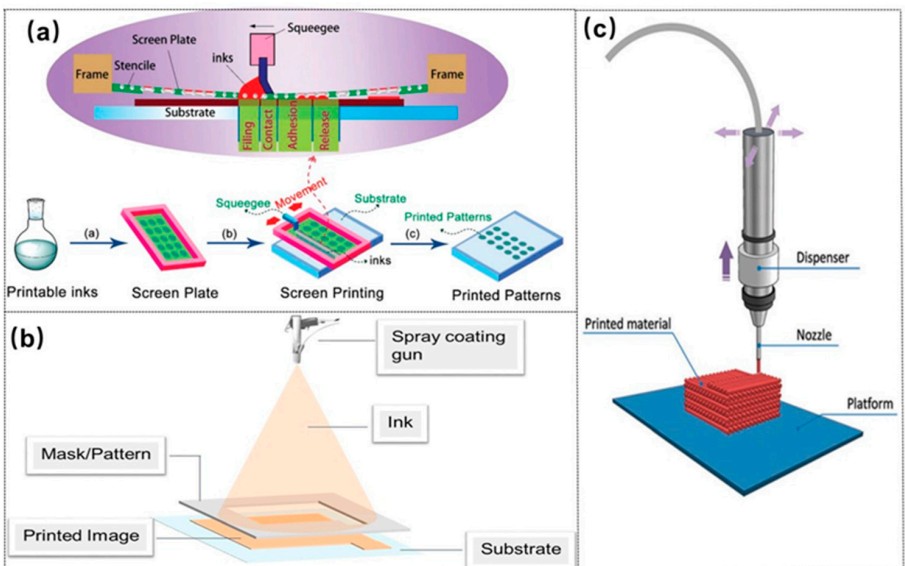

**Figure 15.** Schematics of (**a**) a screen printing [71] (Copyright © 2017, RSC Pub), (**b**) spray printing [72] (Copyright © 2015, John Wiley and Sons), (c) direct ink writing processes [73] (Copyright © 2017, John Wiley and Sons).

Screen printing is one popular and simple method among printing technologies. During screen printing, the ink is pressed through a patterned screen onto the substrate using a roller and forms a film with the structures defined by the patterned screen. Spray printing injects particles from a solution and can easily fabricate large area sheets with a non-contact mode. Meanwhile, direct ink writing using a nozzle can feasibly form a 2D–3D structure with a certain thickness on the substrate. Therefore, different printing technologies can be applied to various battery electrodes.

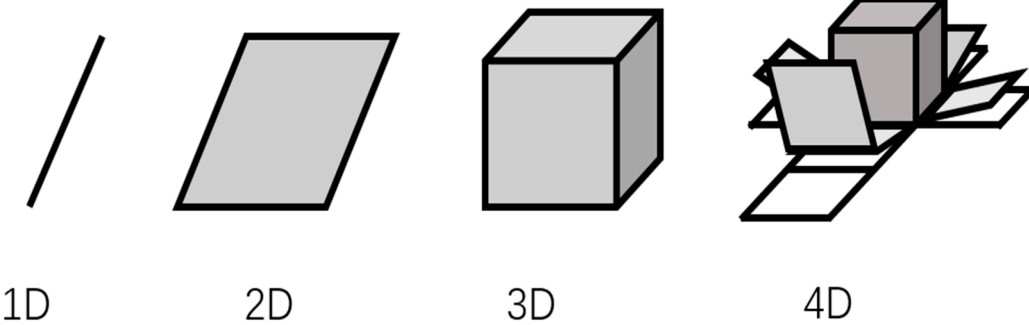

**Figure 16.** Of 1-, 2-, 3-, and 4D concepts. A 4D structure is a structure (x, y, z) made by 3D changes over time (t). Arrows indicate the direction of change with respect to time.

### 5.1. Spray Coating and 2D Printing

Spray coating is a traditional coating method to fabricate composites by depositing particles onto the substrate surface. According to the working mechanism, spray can be divided into two types, cold spray and thermal spray. Thermal spray delivers melted metal drops or non-metal particles at high temperatures and forms a coat on the substrate [74]. Thermal spray has relatively wide applications in metal or alloy materials processing, such as surface coating and corrosion resistance. In contrast to thermal spray, cold spray without heating is a coating process to accelerate particles using the supersonic driving gas passing through a convergent–divergent nozzle and subsequently ejected onto a substrate in high speed [75]. Cold spray enables the delivery of various materials, including high melting point metal materials, low melting point polymer materials, even biomaterials. Helfritch [76] reviewed 24 new applications of cold spray, such as medical devices, electronics, microdevices, and so on.

Printing is another advanced manufacturing method, including inkjet printing, lithography, 3D printing, 4D printing. In this section, we will first discuss lithography and inkjet 2D printing. Inkjet printing [77,78] is additive manufacturing and appears after screen-printing and spin coating. The principle of inkjet printing consists of five stages: drop ejection, drop flight, drop impact, drop spreading, and drop solidification. Inkjet printing has been used in depositing functional inks onto various substrates for numerical devices, to specific, sensors, micro-batteries, solar cell, and other conductive parts of cells [79–82]. Lim [83] thoroughly reviewed technology issues and influence on different substrates for printed capacitive sensors. Furthermore, since Mirkin [84] reported "Dip-pen nanolithography" (DPN) in science, lithography became the focus of contemporary microfabrication. A DPN system is composed of an atomic force microscope tip as a "nib", solid-state substrate as "paper", and molecules with a chemical affinity for the solid-state substrate "ink". By controlling the AFM tip, the nib directly writes controlled patterns on the substrate materials. Lithography contributes to microfabrication in nanomaterials and micro-devices, such as micro-reactors and sensors, micro-optical system [85–88]. Shao [89] reported nanoimprint lithography in the processing of flexible electronics, conductive electrodes, optoelectronic devices, flexible microlens, and flexible sensors. Certainly, it is feasible to print electrodes with a thin-film structure for metal–air batteries.

### 5.2. Laser Processing

Laser processing has gained more and more attention in recent years. Laser ablation, laser cutting, laser welding, laser sintering, laser direct writing, and other laser-assisted synthesis process are powerful tools for precise manufacturing [90,91]. For microfabrication, laser ablation is used to fabricate porous graphene and graphene quantum dots. Laser power, spot size diameter, hatch distance, scanning speed, wavelength, had an influence on the formation of nanomaterials and nanostructures [92,93]. Lasers can be applied for sintering various materials including metals, ceramics, and polymers [94–96]. In recent years, laser processing has been employed in the fabrication of electrodes [97], supercapacitors [98,99], even full batteries [100]. Successful micromanufacturing includes laser-drilling of microholes in LiFePO$_4$ cathode for Li-ion batteries [101] and laser carbonization anode (graphene) for an interdigital film battery [102]. Li [6] reported femtosecond laser-reduced nano joined graphene oxide/Au conductive network as micro-supercapacitors electrodes. Pröll [103] reported femtosecond-laser structuring of LiMn$_2$O$_4$ composite cathodes for Li-ion micro-batteries.

Yu [3] reported laser sintering of printed anodes for AABs. Results indicated that laser sintering can remove the organic solvent from the printed Al nanoparticle slurry and increase the conductivity of the printed anode. Electrochemical characterization demonstrated laser power of 10 W for sintering for better performance, and 3-layer printed anode with a bigger discharge capacity. A 3-layer battery cell can yield a 239 mAh g$^{-1}$ discharge capacity at an operation voltage of 0.95 V, as shown in Figure 17.

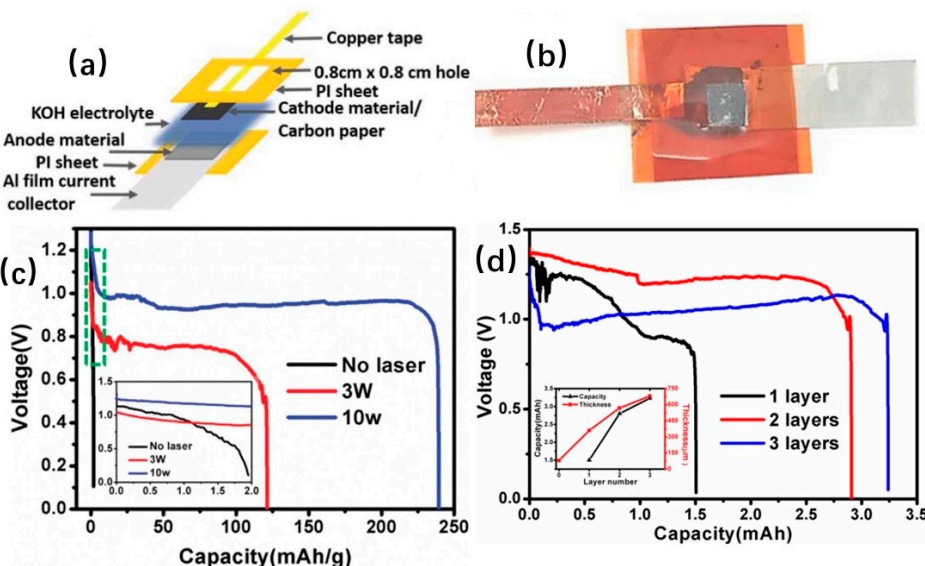

**Figure 17.** (**a**) Schematic of aluminum–air batteries (AAB) working principle, (**b**) Photo of a packaged battery cell, (**c**) The first cycle discharge capacity with different laser sintering powers. (**d**) Discharge capacity for 3D printed anode and relationship between anode thickness and capacity [3] (Copyright © 2018, Electrochemical Society).

### 5.3. 3D Printing

Traditional thin film 2D batteries have suffered from limited energy capacity. 3D printing of flexible micro-batteries with nanostructures can overcome this weakness. Currently, the printed parts of batteries can be electrodes, current collector, solid-state separator, and catalyst in metal–air batteries [104]. Zhou [105] reviewed 3D printing energy storage devices with a sandwich-type and in-plane architecture and demonstrated that the electrochemical energy storage systems can be greatly promoted with 3D printing. Lewis et al. [106,107] reported 3D fully printed electrodes for Li-ion batteries. Li$_4$Ti$_5$O$_{12}$ (LTO) and LiFePO$_4$ (LFP) were separately employed as anode and cathode materials. Electrode material inks were printed onto the substrate, forming multilayer electrodes and an anode and cathode in an interdigitated structure. The results showed that the charge and discharge of 8-layer full cell delivered

1.2 mAh cm$^{-2}$ at a rate of 0.5 C. EIS test revealed the thicker electrode had higher resistance. Meanwhile, CV testing showed the thin wall displayed broader redox peaks. Furthermore, both thin and thick electrodes exhibit excellent Coulombic efficiencies. 3D printed LIBs had 4.45 mAh cm$^{-2}$ at 0.14 mA cm$^{-2}$, corresponding a full cell delivering 14.5 mAh cm$^{-2}$ at 0.2 mA cm$^{-2}$. The same printed technology can also be employed for MABs. For MABs, screen printing has been used in catalysts [108]. A remarkable shortcoming is the narrow choice of suitable materials for printing. In addition, expensive equipment also limits application. However, 3D printing is the destructive technology in MAB manufacturing due to unprecedented designing freedom, high precision, and cost-effective processing.

## 6. Summary and Outlook

In summary, this paper briefly reviewed the recent advances in the studies of the metal–air batteries. Better batteries should be an excellent combination of cathode, anode, and electrolyte., however, there are still some problems to be solved, such as anode side reaction, impure gas $CO_2$ release, electrolyte instability, and so on.

Essentially, improving ORR and OER are quite important to the cathode. Crystallographic structure, materials size, materials morphology, carbon-based materials-doped, and composites can influence different activities of catalyst, which is required to improve both ORR and OER in the cathode.

Nanocomposites and doped-carbon materials are good choices for catalysts in ORR and OER. Research indicates that compared with traditional alloy as an anode, alloys with nanocomposites can reduce the side reaction and improve discharge capacity. While various electrolytes have different advantages more efficient solid-state electrolyte is required in rechargeable metal–air batteries.

Integration of advanced manufacturing, especially 3D printing and laser processing, opens new horizons for MABs. These manufacturing processes allow a better strategy for the systematic combination of the best performance of anodes, cathodes, and electrolyte for improved energy density, efficiency, and cycling stabilities. Although many issues still exist, the further development of MABs, as a compelling alternative to LIBs, holds great promise to address emergent needs of portable electronics, electrical vehicles, and IoTs.

**Author Contributions:** C.W. is responsible for the reference survey, analysis, figure preparation and manuscript drafting. Y.Y. contributes to the laser processing section. J.N., X.L., Y.Z. contribute to the section of "in situ characterization using an electron microscope". Y.L., D.B., J.P. are responsible for the revision of the paper. A.H. advises the review structures, revises the logic order of the chapters and polishes the final manuscript.

**Funding:** This work is partially supported by the National Natural Science Foundation of China (51575016) and NSFC-DFG joint project (51761135129).

**Conflicts of Interest:** The authors declare no conflict of interest.

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
