# Peer review of "Recent Progress of Metal–Air Batteries—A Mini Review"

_applsci, doi:10.3390/app9142787_

Round 1

Reviewer 1 Report

This paper reviews the recent advances on the studies of the metal-air batteries. The paper is well written and structured. The reviewer has the following comments:

In this paper many abbreviations are used. A list of all abbreviations will be helpful for the readers.

The paragraph ‘Flexible batteries’ 122-126 can be extended as this technology currently is gaining interest.

In line 139 your statement is that possible solutions will be introduced in the following section. The suggested solutions can be stated better.

In paragraph ‘Gas diffusion layer’ (154-160) more about the hydraulic micro-channels, their parameters and even some photos could be added.

The results given in lines 210 – 215 could have been better explained. 

Interesting test results are given in paragraph 248-259. Your work would be much interesting for the readers if you manage to present additional information about the better suitability of 4 M NaOH.

In paragraph 261-262 results from literature [52] are presented, according to which Al has lower resistance than Al-Cu alloy, but Al-Cu decreases the anodic potential and improves the battery performance. Additional information about the influence of the bigger Al-Cu resistance might be added.

Line 271, would you quantify that significant improvement.

In chapter ‘Advanced manufacturing of MABs’, a short comparison between screen printing, spray printing and direct ink writing processing would be appreciated by the readers.

Some electrical characteristics under charge and discharge also can be included.

The reviewer assumes that the paper can be published with minor corrections.

Thank you.

Reviewer 2 Report

It was my pleasure to review work: “Recent progress of metal-air batteries – a mini review”. This manuscript is well and thoughtfully  prepared and. It is a solid work which introduce the reader to the subject of metal-air systems for energy storage. Language is consistent and understandable, which makes it easy to follow and broaden reader knowledge.  My few remarks do not change my overall positive impression about this article:

1)      Introductive part can be improved. It should focus on comparison of MABs (or LABs) with LIBs and other energy storage devices. For example there is a completely new brand of energy storage devices which is omitted; lithium-ion capacitors (LICs) which combine two energy storage mechanisms in one device . I provided  a recent reference about LIC so You could read and improve the introduction part:

a.       Nat Mater. 2018 Feb;17(2):167-173

b.       Journal of Materials Chemistry A 4 (32), 12609-12615

c.       Electrochimica Acta 206, 440-445

2)      The uniqueness of MABs is not underlined enough. I think this should be highlighted in the introduction.

3)      Paragraph 5 should be improved. In previous paragraphs You were very precise about all of the details and how they influence the performance of the MAB. Paragraph 5 is more or less mentioning what techniques are used but not why they are used what are the pros and cons?

4)      Paragraph 6 should show more of Your insight. How do You see the evolution of MAB ? What is the crucial advantage and disadvantage? What can be overcome in next few years?

5)      There are some minor orthographic mistakes as in line 373 “ironic liquid” and editorial mistakes. Please read the text very carefully as it is important, especially for the review article.

6)      Representation of the references is not uniform.

Overall, I believe this work after few cosmetic changes mentioned can be accepted for publishing.

Reviewer 3 Report

The mini-review presented in this paper about MAB batteries is a good literature reference for readers with a good background in battery field but also for researchers at an initial stage. Specially intended for this non-expert audience it could be useful a description about the functions and purpose of each component forming part of the fuel cell. A brief description, just about 2 lines.

Also the authors mentioned several the HER, but an explanation about where does it come this hydrogen and the effects on battery performance should be added to introduce such phenomenon.
